# Quality of Life Outcomes in Frontal Sinus Surgery

**DOI:** 10.3390/jcm9072145

**Published:** 2020-07-08

**Authors:** Christos Georgalas, Marios Detsis, Ioannis Geramas, Dimitris Terzakis, Andreas Liodakis

**Affiliations:** 1Department of Surgery—Head and Neck, Medical School, University of Nicosia, Nicosia 2408, Cyprus; cgeorgalas@gmail.com; 2Endoscopic Sinus and Skull Base Surgery—Athens (ESA), Hygeia Hospital, Erythrou Stavrou 4, Maroussi, 15123 Athens, Greece; ioannisgeramas@gmail.com (I.G.); terzodimitris@gmail.com (D.T.); 3Faculty of Mathematics, School of Medicine, European University Cyprus, Diogenis Str 6, Nicosia 2404, Cyprus; mariosdetsis@gmail.com

**Keywords:** frontal sinus surgery, Draf 3 procedure, chronic rhinosinusitis, Draf 2 procedure, patient outcomes, SNOT-22 questionnaire

## Abstract

Introduction: Although significant experience has been gained in the technical nuances of endoscopic sinus surgery procedures, the patient-reported outcomes of frontal endoscopic sinus surgery procedures are still poorly understood. In this study we used the validated patient outcome measure Sino Nasal Outcome Test-22 (SNOT-22) to assess the preoperative and postoperative quality of life in patients undergoing extended endoscopic frontal sinus surgery (Draf type 2 and Draf type 3 procedures). Methods: Out of a total of 680 patients undergoing endoscopic sinus and skull base surgery and 186 patients undergoing frontal sinus surgery, 99 chronic rhinosinusitis patients with (CRSwNP) or without (CRSnNP) nasal polyps undergoing Draf 2 or Draf 3 were assessed. Results: The mean preoperative SNOT-22 was 45.6 points for patients undergoing Draf 2 and 59 for patients undergoing Draf 3, while the mean radiological Lund–Mackay Score was 14.3 and 14.5, respectively. Mean SNOT 22 improvement was 22.9 points for Draf 2 and 37 points for Draf 3 respectively and remained significant in all time intervals, including at 4 years after surgery. With the exception of loss of smell/taste, all symptoms improved by a far bigger extent in Draf 3 group, despite the considerably worse starting point. Effect size (Cohen / Standard Deviations) of Draf 3 was greatest in the following symptoms: “being frustrated/restless/irritable” (1.63), “nasal blockage” (1.43), “reduced concentration” (1.35), “fatigue” (1.29) “runny nose” (1.26) and “need to blow nose” (1.17). Frontal sinus (neo) ostium was patent (fully or partly) at last follow up in 98% of Draf 2 patients and in 88% of patients following Draf 3. Patients with non-patent frontal (neo-) ostium however had a mean postoperative SNOT 22 score of 43 compared to 20 of those with patent frontal sinus (neo-) ostium, although the difference was not statistically significant. Conclusion: Patients undergoing Draf 3 have a greater burden of disease, including both nasal and emotional/general symptoms compared to Draf 2 patients; surgery results in improvement in both groups, although Draf 3 patients have the greatest benefit, especially in emotional / general symptons. In this way both groups achieve similar postoperative quality of life, despite the different starting points.

## 1. Introduction

Chronic rhinosinusitis (CRS) is among the most common diseases in the western world, and its incidence is growing. All patients suffer from impaired quality of life, but especially those who fail medical therapy report health utility values that are significantly lower than the population norm [1]. The primary goal of endoscopic sinus surgery is to improve the quality of life (QOL) of patients who have failed medical therapy [2] as well as to prevent complications and potentially to alter the natural course of the disease [3].

The presence of cardinal symptoms of CRS (nasal obstruction, nasal discharge, face pain or pressure, hyposmia or anosmia) for more than 12 weeks with or without endoscopic and radiological confirmation and resistance to appropriate medical therapy are surgical indications [4]. When affecting the frontal sinuses, chronic sinusitis refractory to medical therapy is an indication frontal sinus surgery.

The first comprehensive classification of endoscopic frontal sinus approaches is credited to Wolfgang Draf, described in 1991 as “the Fulda concept” [5].

Type 1 drainage is established by ethmoidectomy including the cell septa in the region of the frontal recess and excluding the inferior part of Killian’s infundibulum and its mucosa. Resecting the floor of the frontal sinus between the lamina papyracea and the middle turbinate (type 2a) or the nasal septum (type 2b) anterior to the ventral margin of the olfactory fossa after ethmoidectomy produces extended drainage. Type 3 further enlarges the 2b opening by resecting the superior anterior nasal septum and drilling the floor of the frontal sinuses, between the orbits and up to the first olfactory fibers [6,7]. The choice of the appropriate procedure depends on the disease (recalcitrant, Samter’s triad), the anatomy (anteroposterior and lateral dimensions of frontal recess) and the patient (primary or revision surgery, other comorbidities,); however, the general principle of frontal sinus surgery applies: the aim is to create durable openings for sufficient drainage of frontal sinus and for improved delivery of topical therapies [8].

Since its validation from Hopkins and colleagues, Sino Nasal Outcome Test – 22 (SNOT-22) has become the reference questionnaire to assess health status and health-related quality of life in CRS [9]. It originated from the Rhinosinusitis Outcomes Measure-31 (RSOM-31), developed by Piccirillo and co-authors. The RSOM-31 contains 31 RS-specific items (e.g., runny nose, cough, facial pain/pressure), grouped into 7 domains (nasal, eye, sleep, ear and general symptoms; practical problems and emotional consequences), and was created from discussions with chronic sinusitis patients [10]. The need for simpler questionnaires gave rise to SNOT-20, where 11 items of RSOM-31 were removed, and the addition of two cardinal symptoms of CRS (“stuffed nose” and “difficulty to smell or taste” [4]) led to the final form of SNOT-22 [11].

The goal of this paper is to assess the effect of endoscopic frontal sinus surgery (Draf 2 and Draf 3) on health-related quality of life, as recorded in pre- and postsurgical SNOT-22 questionnaires. A secondary goal was to assess the correlation between endoscopic findings (neo-ostium patency) and patient reported QOL as well as with radiological disease severity as recorded by Lund–Mackay grading scale.

## 2. Materials and Methods

Out of 670 patients undergoing endoscopic sinus surgery and a total of 186 patients having undergone endoscopic frontal sinus surgery (Draf 2 or Draf 3) for a variety of diagnoses (sinonasal malignancies, skull base tumors, cerebrospinal fluid repair) by the first author (CG) over the past 5 years, we selected 99 patients where the indication for surgery was chronic rhinosinusitis and who underwent frontal sinus surgery in isolation. We excluded (a) all patients where more sinus procedures in addition to frontal sinus surgery were performed and (b) all cases where the indication for frontal sinus surgery was not chronic rhinosinusitis (benign or malignant sinonasal or skull base tumors, CSF leak repair, complications of acute rhinosinusitis such as Potts Puffy tumor, frontocutaneous fistula, etc.) We conducted a retrospective study of these 99 patients who underwent Draf 2 and Draf 3 procedures by the first author (CG) from 2015 to 2020, assessing preoperative SNOT-22 scores and comparing preoperative with postoperative scores. All 99 patients suffered from CRS with (CRSwNP) or without (CRSsNP) nasal polyps and underwent frontal sinus surgery as a sole sinus procedure. The specific type of Draf procedure was decided preoperatively, according to the duration and severity of symptoms, number and type of previous operations, presence of comorbidity (such as Samter’s triad, allergy, asthma, etc.) and anatomical factors (anteroposterior and lateral size of frontal recess and presence of frontal cells). Patients listed for frontal sinus surgery were those resistant to appropriate medical treatment, as defined in International Consensus Statement in Allergy and Rhinology—RhinoSinusitis—2016 (ICARS 2016) guidelines [4] (following nasal rinsing, topical steroids, oral steroids and/or oral antibiotics as needed) while concomitant allergies were assessed and treated. We recorded the preoperative computed tomography (CT) findings using Lund–Mackay scale (L-M) [12]. As per L-M, we graded each frontal sinus as 0—completely clear, 1—partly opacified, 2—completely opacified (total score for both frontal sinus 0–4 and total score for all sinuses 0–24). Postoperative care included follow-up appointments at 1 week, 3 weeks, 6 weeks and 3 months postoperatively for gentle nasal debridement. All patients were advised to use high-volume nasal irrigations with topical budesonide from the second postoperative day for at least 3 months. All patients with CRSwNP were given oral steroids for 10 days postoperatively, and those where purulent secretions were present were given 10 days of oral, culture-targeted antibiotics. All patients had at least one SNOT-22 questionnaire (pre-or postop) conducted. Patients without preoperative or postoperative questionnaires were excluded. When more than one preoperative questionnaire per patient were reported, we only included the most recent one. All postoperative SNOT-22 questionnaires were included in the study, and we divided the postoperative period into 5 terms; 1–3 months, 3–6 months, 6–12 months, 1–2 years and 2–3 years after surgery. All patients with postoperative SNOT-22 questionnaires reported their first one during the first-time interval (1–3 months postoperatively).

Every SNOT-22 questionnaire was reported as a total score and was also analyzed in 22 subscores, each of them corresponding to a single item. We also recorded the postoperative status of the frontal sinus ostium/neo-ostium in all patients as follows: 0—closed, 1—partly open, 2—fully open. In cases of bilateral Draf 2, we calculated the mean of both ostia. The scientific and ethics board of our hospital (Scientific Council of Hygeia Hospital—chair: Dr. G. Zacharopoulos) approved this questionnaire-based outcome study (Project id code: 2019/2—Outcome frontal). All patients provided informed consent to the use of the anonymized SNOT 22 questionnaires and their data for the purpose of this study.

### 2.1. Statistical Analysis

A Linear mixed model to allow both fixed and random effects was utilized to determine the changes of the SNOT22 score before surgery and in different time intervals after the surgery. We choose this model due its suitability in dealing with missing values and because there was correlation within subjects as we had to deal with repeated measures for the same individual. *p* Values < 0.05 were considered statistically significant.

### 2.2. Statistical Software

In the present study, statistical analysis and graphical representation were performed using SPSS Statistics Subscription (2020) and Microsoft^®^ Office Excel 365.

## 3. Results

A total of 99 patients underwent frontal sinus Draf procedures (76 Draf 2 (21 Draf 2a–55 Draf 2b) and 23 Draf 3) between 16 September 2015 and 25 February 2020. The mean age of patients was 45.7 years (range 12 to 80), and 54 of them were male. All patients underwent CT preoperatively with a mean total L-M score of 14.5 (SD 5.4) and frontal sinus radiological score of 2.46 (SD 1.3). The preoperative questionnaire was administered a mean of 19.5 (8–27) days before surgery. Postoperative follow up was divided into 5-time intervals. Eighty six out of 99 (87%) of patients suffered from CRSwNP. There was no correlation between L-M score or frontal sinus radiological scores and SNOT 22 scores (Pearson correlation coefficient: 0.12, *p* = 0.25 and 0.081 and *p* = 0.44 respectively). 

Average follow up was 24 months (range: 5 to 58 months). Characteristics of patients operated are summarized in Table 1. In most patients undergoing Draf 3 local flaps or grafts were used to ensure patency of the neo-ostium. 

There was a highly significant difference in total SNOT 22 score between the patients listed for Draf 2 (mean score 45.6) and those listed for Draf 3 (mean score 59) (*p* = 0.02). This reflected higher symptom burden across the board for those undergoing Draf 3, with the difference in symptoms reaching significance in the areas of **thick nasal discharge, dizziness, reduced productivity, reduced concentration, frustrated/restless/irritable and sadness**. The most troublesome symptoms (as graded preoperatively) included **nasal obstruction, loss of smell/taste and need to blow nose** for Draf 2 group and **loss of smell/taste, thick nasal discharge, need to blow nose** for Draf 3 group, as well as **waking up tired, fatigue, reduced productivity, reduced concentration and frustration-irritability** (Table 2).

This difference in SNOT 22 scores between the two groups disappeared postoperatively (1–3 months post-op): SNOT 22 total scores were 22.7 for the Draf 2 group and 22 for the Draf 3 group (*p* = 0.8). This reflects the greater improvement patients felt after Draf 3 (37 points) compared with Draf 2 (22.9 points). (Figure 1)

### 3.1. Symptom by Symptom Analysis—1–3 Months Postoperative Follow-Up

Specifically, regarding every SNOT-22 item, all symptoms improved significantly in the Draf 2 group, with the biggest effect size in nasal blockage (1.02 Standard Deviations). In the Draf 3 group, all symptoms improved as well but by a far greater margin (with the exception of loss of smell/taste), and that was also the case for the domains of psychological and sleep dysfunction. The effect size of surgery was greatest in the symptom **being frustrated/restless/irritable** (1.63), followed by **nasal blockage** (1.43), **reduced concentration** (1.35), **fatigue** (1.29) **runny nose** (effect size 1.26) and **need to blow nose** (1.17). (Table 3)

Subsequent follow-up: Patients were followed up with multiple SNOT 22 questionnaires at different time intervals. We analyzed the results by time-frame as follows: 1–3 months (* mean 68 days), 3–6 months (mean 155 days), 6–12 months (mean 345 days), 12–24 months (mean 473 days) and 24+ months (mean 912 days) (Table 4)

### 3.2. Endoscopic Examination of Frontal Sinus (Neo)-Ostium

At last follow up, in one Draf 2 patient, the frontal ostium was closed, in 11 (15.5%) it was partly open, and in 59 (83%), it was fully open. In Draf 3 patients, one neo-ostium was completely obstructed, 9 (31%) were partly open, and 13 (56.5%) were completely open. Snot 22 scores were worse for patients with closed frontal sinus ostium (mean 43) or partly open (Mean 21.5) than for those where it was fully open (19), although the difference was not statistically significant. (Figure 2)

## 4. Discussion

Frontal sinus surgery has evolved from radical, highly invasive, disfiguring approaches to function-preserving, minimally invasive and non-disfiguring intranasal procedures. Most sinus surgeons would agree that a successful surgical procedure is one that improves patient’s symptoms and prevents complications [13].

The development of endoscopic sinus techniques has rendered osteoplastic frontal sinus obliteration (OFSO) irrelevant for the vast majority of patients with chronic sinusitis. Hence, there are no data assessing quality of life outcomes with SNOT-22 questionnaires after open techniques. A study published by Joshua B. Silverman et al. in 2012 reviewed 39 patients who underwent OFSO, where the outcome of surgery was assessed by symptom resolution through a patient self-filled questionnaire and the need for revision surgery. A rate of 91% improvement was reported; however, the questionnaire used was not validated [14].

### 4.1. Endoscopic Sinus Surgery

We set up this study in order to compare patients undergoing Draf 2 and Draf 3 procedures. Using strict exclusion criteria, we included exclusively patients suffering from CRSs/wNP. Out of 670 patients undergoing endoscopic sinus surgery and a total of 186 patients having undergone endoscopic frontal sinus surgery (Draf 2 or Draf 3) for a variety of diagnoses (sinonasal malignancies, skull base tumors, cerebrospinal fluid repair) over the past 5 years, we selected 99 patients where the indication for surgery was chronic rhinosinusitis and who underwent frontal sinus surgery in isolation. In an earlier study [15], we have shown that there is a poor correlation between QOL and radiological grading; in this study, we showed the same—although radiology is crucial for planning surgery and estimating the extent of disease, it is a poor predictor of QOL impairment (or improvement). Indeed, we found a Pearson correlation of less than 0.2 between L-M score and preoperative SNOT 22.

Both groups of patients, especially the patients of Draf 3 group, suffered from moderate to severe disease and reported high scores in preoperative SNOT-22 questionnaires; **the mean preoperative SNOT-22 score was 45.6 for Draf 2 group and 59 for Draf 3 group (*p* = 0.02)**. These scores were higher in comparison with patients undergoing endoscopic sinus surgery in general. A recent cohort study by Rabii Laababsi et al. divides CRSnNP patients undergoing FESS (Functional Endoscopic Sinus Surgery) into two groups, according to the extent of disease (unilateral or bilateral), and the mean preoperative SNOT-22 scores were 37.13 and 41.76, respectively [16]. Moreover, in a large study of 2263 CRS patients undergoing FESS from 87 UK hospitals, preoperative SNOT-22 scores had a near normal distribution around a mean of 42.5 [17]. Hence, it appears that patients suffering from frontal sinus disease (and indeed our sample) have a higher burden than the general population of CRS patients undergoing FESS.

Regarding the effect of surgery, the study of Zachary Soler et al. [18] included 40 unique patient cohorts published from 2008–2016. All studies showed a statistically significant change in mean SNOT-22 scores between baseline and post-operative time points and the summary change in mean SNOT-22 across all studies was 24.4 points. According to Systematic Review and Meta-analysis of SNOT-22 Outcomes after Surgery for Chronic Rhinosinusitis with Nasal Polyposis, which was published in 2018 by Phong T. Le et al. [19], a mean change of 23.0 points was the result of pooled analyses of SNOT-22 scores reported in fifteen articles with 3048 patients. In our study, the effect of Draf 2/Draf 3 surgery is more robust, producing an improvement of **37 points in SNOT 22 score for patients undergoing Draf 3 and of 23 points for those undergoing Draf 2 (*p* < 0.0001)**. Both of these systematic reviews suggested that worse preoperative SNOT-22 scores were accompanied by a larger change in SNOT-22 scores, and our results are consistent with this conclusion. A mean decline of 21.5–24.3 points in postsurgical SNOT-22 scores is reported by Daniel M. Beswick et al. in a comparison of surgical outcomes between 190 patients with unilateral and bilateral chronic rhinosinusitis without nasal polyps, with statistically not significant more improvement in patients with bilateral disease [20]. The mean follow-up time was 15.4–16.2 months and mean age at presentation was 51.8–53.0 years. In our case, mean follow-up was longer (average of 24 months) and mean age was 43 and 47 years, respectively.

### 4.2. Frontal Sinus Surgery

We reported significantly worse preoperative SNOT-22 total scores in Draf 3 group as well as worse subscores regarding ear/facial symptoms, sleep disturbance and psychological dysfunction. These results validate our selection criteria: The subgroup of patients undergoing Draf 3 are different not only in medical terms (number of previous operations, comorbidity, allergy, etc.) but, importantly, report a much higher burden of disease, which seems to affect every part of their life. **Fatigue, reduced concentration and irritability** score as high as nasal symptoms—reflecting the psychological burden of recalcitrant sinusitis. Crucially, these symptoms are the ones to show the biggest improvement after Draf 3 and result in the much bigger benefit in QOL seen with Draf 3 compared to Draf 2. The effect size of surgery was greatest in the symptom **being frustrated/restless/irritable** (1.63), followed by **nasal blockage** (1.43), **reduced concentration** (1.35), **fatigue** (1.29) **runny nose** (effect size 1.26) and **need to blow nose** (1.17). As a result, patients selected for Draf 3, end up with similar QOL to those initially selected for Draf 2, despite starting from a much worse baseline, vindicating the choice of a more aggressive procedure.

Comparing the quality of life outcomes between Draf 2 and Draf 3 procedures, we found a greater and statistically significant mean change in SNOT-22 scores at follow up vs preoperatively in Draf 3 group. The study of Patel V.S. et al. published in 2018 reveals a similar weak correlation. In this study, mean change in SNOT-22 scores for Draf 2b and Draf 3 scores were 24.1 and 24.9, respectively, at last follow-up vs preoperatively. The Draf 2b group had greater improvement in SNOT-22 score than the Draf 3 group at 1 to 3 months (*p* = 0.003), but the magnitude of improvement equalized at 5 to 9 months (*p* = 0.66) and last follow-up (*p* = 0.90) [21]. Further data suggesting poorer quality of life outcomes of Draf 3 in early postoperative time is reported by Jafari A. et al. in a retrospective study of 19 patients. In comparison to a control group of 19 Draf 2 patients, the mean SNOT-22 scores improved for both groups (−12.7 ± 34.5 versus −9.5 ± 20.4; *p* = 0.74) over the follow-up period, and subscore (domain) analysis demonstrated worsening extranasal symptoms (2.5 ± 3.0 versus −1.5 ± 4.7; *p* < 0.05) at the first postoperative visit and less improvement in ear/facial symptoms at the second (−0.5 ± 2.6 versus −3.9 ± 4.7; *p* = 0.03) and third postoperative visits (−1.1 ± 1.6 versus 3.5 ± 3.3; *p* = 0.01) [22]. That kind of short-term morbidity after the Draf 3 procedure is not supported by our study, where all symptoms regardless the surgery technique appear to be significantly improved from the first postoperative visit (1–3 months postoperatively), with the exception of loss of smell/taste in Draf 3 group. Olfaction is reported improved at first postoperative follow-up in Draf 3 group but without statistical significance. We also reported ear, facial and “psychological” symptoms’ relief early during follow-up period with much bigger improvement in Draf 3 group.

Regarding the subsequent follow up, we reported higher total SNOT-22 scores in both groups, but with much less questionnaires reported while the overall improvement in SNOT 22 across the board continues, albeit with a small worsening. It is possible that this “worsening” reflects the more persistent cases; patients with persistent symptoms tend to be stricter with their postoperative visits, whereas patients without annoying symptoms are, in general, more difficult to keep in touch.

A multi-institutional cohort study comparing the effectiveness of surgical (273 patients) vs medical (69 patients) management of chronic sinusitis was published by DeConde et al. in 2015. Beginning from a worse baseline, all patients who underwent endoscopic sinus surgery, showed much more improvement in thick nasal discharge (odds ratio [OR] = 4.36), facial pain/pressure (OR = 3.56), and blockage/congestion of nose (OR = 2.76) [23]. Regarding olfaction, a review of 104 patients undergoing Draf 3 was published in 2019 and the mean change in olfaction-subscore was 1.54 points, where the mean change in the same subscore in our study was measured at 0.87 points at first follow-up [24].

Comparing the quality of life outcomes between men and women, we found identical preoperative SNOT-22 scores in men and women. This is inconsistent with other QOL measures, where often women report worse quality of life scores, although their endoscopic and radiological appearance may be better than that of men [25] and it likely reflects the “objective” significant symptomatic severity of the disease in our patients undergoing Draf 2 and Draf 3 procedures. Moreover, the mean change in SNOT-22 scores at follow-up vs preoperatively and the magnitude of improvement seem to be similar between the two genders, a hypothesis that is strongly supported by Lal D. et al. [26]. We did not find a statistically significant correlation between status of neo-ostium and SNOT 22, reflecting our 2011 study in long term results of Draf 3 [7]), although this could be related to the small number of patients with closed neo ostium. However, there was a trend towards worse QOL (from 43 to 20) in patients with closed neo-ostium. This study has some limitations: Not all patients had both pre- and postoperative questionnaires; however, we feel that this did not affect the validity of analysis, as we did not compare results by performing paired t-tests. We did have long-term outcome results for all patients, and one could argue that results worsen by time. However, this is not our feeling; we think it is always hard to keep track of patients followed up for a benign disease, and it is likely that only patients with significant symptoms would turn up for follow-up after 2 or 3 years.

## 5. Conclusions

We reviewed 99 patients undergoing frontal sinus surgery by the forst authors for CRSw/s over a five-year period. Patients undergoing Draf 3 have a greater burden of disease, including both nasal and emotional/general symptoms compared to Draf 2 patients; surgery results in improvement in both groups, although Draf 3 patients have the greatest benefit, especially in emotional/general symptoms. In this way both groups achieve similar postoperative quality of life, despite the different starting points. 

In carefully selected patients with CRSw/sNP, Draf 3 procedure offers a very significant improvement in their quality of life, including both nasal and generic symptoms. Further investigation with SNOT-22 questionnaires as gold standard measure of quality of life is essential for a more definite answer to “who” and “when” a patient benefits from endoscopic frontal sinus surgery.

## Figures and Tables

**Figure 1 jcm-09-02145-f001:**
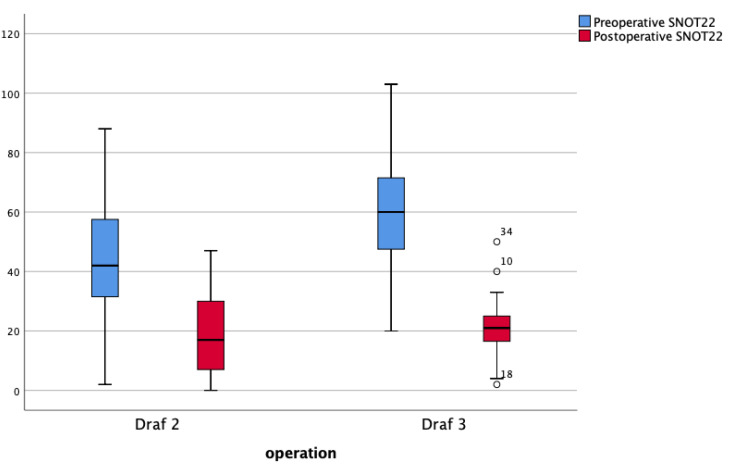
Preoperative and postoperative median SNOT 22 (Sino Nasal Outcome Test-22) scores by operation.

**Figure 2 jcm-09-02145-f002:**
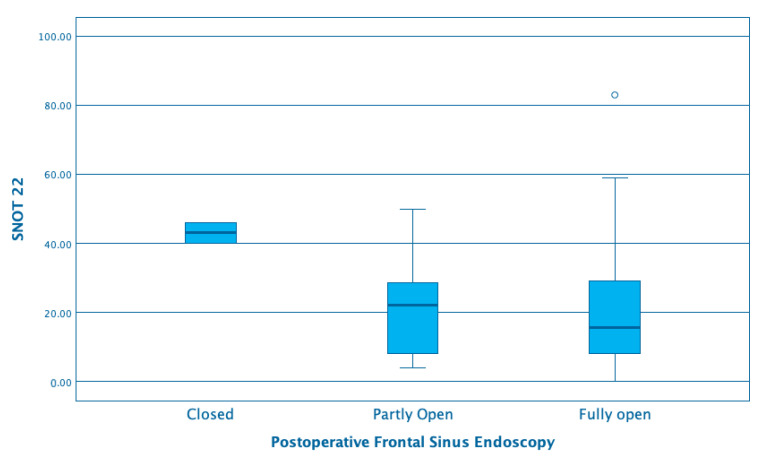
SNOT 22 mean and range by postoperative frontal sinus endoscopy.

**Table 1 jcm-09-02145-t001:** Descriptive statistics per Draf procedure.

	Draf 2a (*N* = 21)	Draf 2b (*N* = 55)	Draf 3 (*N* = 23)	*p*
Mean age/range	43.8 (19–68) years	47.5 (12–80) years	43 (17–61) years	ns
Lund–Mckay CT Score: mean(SD)	14.1 (5.1)	14.6 (5.3)	14.9 (5,8)	ns
Frontal sinus CT score:	2.4 (1.3)	2.3 (1.4)	2.5 (1.2)	ns
Male/female ratio	13/8	27/28	14/9	ns
Revision surgery	0	1	4	ns
Local flap	0	1	17	*p* < 0.01
Septoplasty and/or turbinoplasty	4	7	0	ns
Asthma	0	0	1	ns
AERD	0	1	2	ns

Preoperative: Total SNOT 22 and symptom by symptom analysis (CT—Computed Tomography, SD—Standard Deviation, AERD—Aspirin Exacerbated Respiratory Disease, ns—not significant).

**Table 2 jcm-09-02145-t002:** Preoperative grades per symptoms—included in SNOT-22 questionnaire. Comparison between Draf 2/Draf 3.

	Operation	Mean	*p*-Value
**SNOT22**	**Draf 2**	**45.63**	**0.023**
	**Draf 3**	**59.00**	
Need to blow nose:	Draf 2	2.91	0.104
	Draf 3	3.63	
Sneezing:	Draf 2	1.58	0.081
	Draf 3	2.31	
Runny nose:	Draf 2	2.03	0.050
	Draf 3	2.94	
Nasal blockage:	Draf 2	3.56	0.996
	Draf 3	3.56	
Loss of smell/taste:	Draf 2	3.02	0.087
	Draf 3	3.94	
Cough:	Draf 2	1.42	0.572
	Draf 3	1.69	
Postnasal drip:	Draf 2	2.62	0.406
	Draf 3	3.00	
**Thick nasal discharge:**	**Draf 2**	**2.68**	**0.029**
	**Draf 3**	**3.69**	
Ear fullness:	Draf 2	1.61	0.273
	Draf 3	2.13	
Dizziness:	Draf 2	1.23	0.034
	Draf 3	2.13	
Ear pain:	Draf 2	0.85	0.791
	Draf 3	0.94	
Facial pain or pressure:	Draf 2	2.30	0.584
	Draf 3	2.56	
Difficulty falling asleep:	Draf 2	2.12	0.523
	Draf 3	2.44	
Wake up at night:	Draf 2	2.20	0.785
	Draf 3	2.06	
Lack of a good night’s sleep:	Draf 2	2.56	0.703
	Draf 3	2.75	
Wake up tired:	Draf 2	2.62	0.262
	Draf 3	3.13	
Fatigue:	Draf 2	2.52	0.273
	Draf 3	3.00	
**Reduced productivity:**	**Draf 2**	**1.92**	**0.012**
	**Draf 3**	**3.13**	
**Reduced concentration:**	**Draf 2**	**1.74**	**0.004**
	**Draf 3**	**3.00**	
**Frustrated/restless/irritable:**	**Draf 2**	**2.11**	**0.008**
	**Draf 3**	**3.31**	
**Sadness**:	**Draf 2**	**1.17**	**0.012**
	**Draf 3**	**2.13**	
Embarrassment:	Draf 2	0.83	0.056
	Draf 3	1.56	

In bold the statistically important differences between the two groups.

**Table 3 jcm-09-02145-t003:** Draf 2/Draf 3 group-mean change in every SNOT-22 item at postoperative follow-up at 1–3 months.

	Draf 2			Draf 3		
	Mean Change	Standard Deviation	*p* Value	Mean Change	Standard Deviation	*p* Value
SNOT22	22.87	20.9	0.000	37.00	25.73	0.000
Need to blow nose	1.47	1.78	0.000	2.27	1.94	0.000
Sneezing	0.45	1.66	0.048	1.53	1.30	0.000
Runny nose	1.24	1.66	0.000	2.47	1.95	0.000
Nasal blockage	1.89	1.84	0.000	2.33	1.63	0.000
Loss of smell/taste	1.29	2.20	0.000	0.87	2.20	0.149
Cough	0.69	1.39	0.001	0.93	1.53	0.034
Postnasal drip	1.02	1.75	0.000	1.47	1.84	0.008
Thick nasal discharge	0.94	1.99	0.001	1.87	1.72	0.001
Ear fullness	0.62	1.77	0.013	1.33	2.22	0.036
Dizziness	0.54	1.50	0.009	1.73	1.48	0.000
Ear pain	0.40	1.34	0.031	0.80	1.20	0.022
Facial pain or pressure	1.11	1.80	0.000	1.33	1.79	0.012
Difficulty falling asleep	1.45	1.58	0.000	1.67	1.98	0.006
Wake up at night	1.30	1.62	0.000	1.27	2.07	0.027
Lack of a good night’s sleep	1.37	1.47	0.000	1.33	1.92	0.041
Wake up tired	1.27	1.40	0.000	1.80	2.11	0.005
Fatigue	1.05	1.61	0.000	1.60	1.24	0.006
Reduced productivity	0.73	1.72	0.000	1.80	1.30	0.005
Reduced concentration	0.60	1.16	0.008	2.13	1.58	0.000
Frustrated/restless/irritable	1.00	1.02	0.000	2.53	1.61	0.000
Sadness	0.56	20.93	0.001	1.33	25.73	0.006
Embarrassment	0.38	1.78	0.008	1.20	1.94	0.012

**Table 4 jcm-09-02145-t004:** Mean SNOT-22 scores evolution over time. Draf 2/Draf 3.

	1–3 Months	3–6 Months	6–12 Months	12–24 Months	24–48 Months
Draf 2	22.7 (*N* = 55)	20.9 (*N* = 37)	25.5 (*N* = 18)	29.5 (*N* = 9)	31.0 (*N* = 2)
Draf 3	22 (*N* = 18)	21.2 (*N* = 14)	31.0 (*N* = 8)	38.5 (*N* = 7)	7.5 (*N* = 3)

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
