# Peer review of "Quality of Life Outcomes in Frontal Sinus Surgery"

_jcm, 2020, doi:10.3390/jcm9072145_

Round 1

Reviewer 1 Report

I consider the paper suitable for publication

Reviewer 2 Report

Revision of manuscript is now adequate. 

This manuscript is a resubmission of an earlier submission. The following is a list of the peer review reports and author responses from that submission.

Round 1

Reviewer 1 Report

Thank you for the opportunity to review this work. The article by Liodakis et al “Quality of Life Outcomes in Frontal sinus surgery” reports the outcomes of 99 cases treated with endonasal endoscopic approach for isolated frontal chronic rhinosinusitis, with a particular focus on SNOT-22 questionnaire changes (preoperatively vs postoperatively).

Despite the paper is interesting and has a lot of worth, the manuscript needs to be revised. The following features need to be addressed:

  • Provide a table reporting the total cases, the Draf 2a cases, the Draf 2b cases and the Draf 3 cases. For each subgroup report mean age and range, male to female ratio, revision surgery and primary surgery, report the number of patients that had local flaps if they have been used to cover the exposed bone, and provide the number of patients that have been submitted to an additional septoplasty and/or turbinoplasty to access the frontal recess and frontal beak areas.
  • I recommend to perform a more extensive analysis of your results, evaluating the SNOT-22 outcomes by comparing different subsets of patients defined by different variables: e.g., patients having had additional septoplasty or turbinoplasty vs those without, patients affected by CRSwNP vs CRSsNP, different ages, comorbidities (e.g., asthma, allergy, etc). This analysis could reveal some relations between surgical outcome (SNOT-22) and to one of these variables.
  • Considering that this is a chronic disease that you are dealing with, a firm endoscopic follow-up of the Draf’s patency is required as a stenosis may occur. In general, this is particularly true for CRSwNP (being the most of your patients). The authors should put in relation the results of the postoperative SNOT-22 with the surgical outcomes, clinically evaluated. Considering that the table with the scores evolution shows a progressive SNOT-22 worsening during the follow-up, an endoscopic correlation that might reveal a stenosis of the Draf should be provided. On this matter, it could be interesting to report symptom-by-symptom SNOT-22 not only at 2 months of follow-up but also at late follow-up. It is my belief that you need to evaluate the long-term surgical outcome when dealing with this disease and these surgical procedures, while it seems that you give more importance to the early postoperative period.
  • In Table 4 you report at 2 months follow-up a SNOT-22 mean score 22.87 for Draf 2 and 37,00 for Draf 3. However, on the next table with the mean SNOT-22 scores over time you report at 1-3 months follow-up a a SNOT-22 mean score 22,8 for Draf 2 and 22 for Draf 3. Also in line 112 you report a SNOT-22 score for Draf 2 as 22,7. Please correct these discrepancies. Also report which is the follow-up time in line 112-113 (2 months post-op). Moreover, on the table with the mean SNOT-22 scores over time, there is a score for Draf 3 that is 7,5 at 24-48 months which is very low compared to the previous score at 12-24 months. Please check this issue and eventually correct or discuss these results.
  • There are some bias regarding the fact that not all patients had preoperative questionnaire and not all patients had postoperative questionnaire. Moreover, there were some patients that have had only one post-op questionnaire that it may have been at the early or at late follow-up. Please discuss these issues/bias.
  • References should be cited following the journal’s reference style.

Author Response

Dear reviewer,

Many thanks for going through our manuscript and for your helpful suggestions. We went through your suggestions and adapted our manuscript accordingly, as follows:

Provide a table reporting the total cases, the Draf 2a cases, the Draf 2b cases and the Draf 3 cases. For each subgroup report mean age and range, male to female ratio, revision surgery and primary surgery, report the number of patients that had local flaps if they have been used to cover the exposed bone, and provide the number of patients that have been submitted to an additional septoplasty and/or turbinoplasty to access the frontal recess and frontal beak areas.

We constructed a new table (Table 2) according to your suggestions.

I recommend to perform a more extensive analysis of your results, evaluating the SNOT-22 outcomes by comparing different subsets of patients defined by different variables: e.g., patients having had additional septoplasty or turbinoplasty vs those without, patients affected by CRSwNP vs CRSsNP, different ages, comorbidities (e.g., asthma, allergy, etc). This analysis could reveal some relations between surgical outcome (SNOT-22) and to one of these variables.”

Due to small amounts of patients with additional septoplasty and/or turbinoplasty (only 11 out of 99) or comorbidities such as asthma and AERD (only 4 patients totally), we found it meaningless to make comparisons regarding these variables. Moreover, patients with CRSwNP were much more than those with CRSnNP (86vs13), a fact that eliminated our ability to make comparisons between them.

Considering that this is a chronic disease that you are dealing with, a firm endoscopic follow-up of the Draf’s patency is required as a stenosis may occur. In general, this is particularly true for CRSwNP (being the most of your patients). The authors should put in relation the results of the postoperative SNOT-22 with the surgical outcomes, clinically evaluated. Considering that the table with the scores evolution shows a progressive SNOT-22 worsening during the follow-up, an endoscopic correlation that might reveal a stenosis of the Draf should be provided. On this matter, it could be interesting to report symptom-by-symptom SNOT-22 not only at 2 months of follow-up but also at late follow-up. It is my belief that you need to evaluate the long-term surgical outcome when dealing with this disease and these surgical procedures, while it seems that you give more importance to the early postoperative period.”

Well, it’s true that we focused on the quality of life outcomes of frontal sinus surgery.  The patency of the opening of the draf 2/ draf 3 is also a valid outcome, but has been examined extensively in a number of other publications. We found (like most other reserachers) that there was a poor correlation between the patency of the Draf opening and the patient quality of life. We agree with importance of long term follow up – however, as you are aware, it is not always easy to keep track of patients followed up for a long period of time for a benign condition. Although there is a risk that symptoms deteriorate, it is also a fact that patients with greater postoperative improvement were less likely to keep on attending for follow up and thus reporting questionnaires, hence we feel it would be quite risky to talk about worsening of quality of life as the postoperative years increase. We have added this as a limitation in our study as follows: “We did have long term outcome results for all patients – and one could argue that results worsen by time: However, this is not our feeling: we think it is always hard to keep track of patients followed up for a benign disease, and it is likely that only patients with significant symptoms would turn up for followup after 2 or 3 years.

In Table 4 you report at 2 months follow-up a SNOT-22 mean score 22.87 for Draf 2 and 37,00 for Draf 3. However, on the next table with the mean SNOT-22 scores over time you report at 1-3 months follow-up a a SNOT-22 mean score 22,8 for Draf 2 and 22 for Draf 3. Also in line 112 you report a SNOT-22 score for Draf 2 as 22,7. Please correct these discrepancies. Also report which is the follow-up time in line 112-113 (2 months post-op). Moreover, on the table with the mean SNOT-22 scores over time, there is a score for Draf 3 that is 7,5 at 24-48 months which is very low compared to the previous score at 12-24 months. Please check this issue and eventually correct or discuss these results.”

Our apologies if it was not clear enough. In Table 5 we reported mean change of SNOT-22 scores at first postoperative follow-up, when in Table 6 we reported postoperative SNOT-22 scores regarding five time intervals. It is true that the mean SNOT-22 score for Draf 3 group at the fifth interval is just 7,5, however it is made out of only 3 patients. We revised the table by mentioning the total number (N) of questionnaires reported at each time interval. We made clear (line 129 and Table 5) that the right first time interval is 1-3 months postop.

There are some bias regarding the fact that not all patients had preoperative questionnaire and not all patients had postoperative questionnaire. Moreover, there were some patients that have had only one post-op questionnaire that it may have been at the early or at late follow-up. Please discuss these issues/bias.”

True, not all patients had both pre and postoperative questionnaires – although all patients had either pre or postoperative   questionnaires. Statistically, this was accounted for, by not performing paired t-tests. We added the following sentence in “Materials and Methods” paragraph: “All patients with postoperative SNOT-22 questionnaires, reported their first one during the first time interval (1-3 months postoperatively).” We also added the sentence in discussion – “Although not all patients had both pre and postoperative questionnaires, this did not affect the validity of analysis, as we did not compare results by performing paired t-tests”.

Reviewer 2 Report

Well written article with a single clarification/modification as listed below.

Structure of manuscript is good.  

Statistical analysis is adequate.

The concept of using SNOT 22 scoring is appropriate.

My only suggestion to the authors is to clarify the preoperative criteria which patient was to undergo a Draf 2 vs a Draf 3 procedure.  

Was this decided prep or during surgery.   It is not clearly stated in the manuscript.

Author Response

Dear reviewer,

Many thanks for going through our manuscript and for your helpful suggestions. We went through your suggestions and adapted our manuscript accordingly, as follows:

My only suggestion to the authors is to clarify the preoperative criteria which patient was to undergo a Draf 2 vs a Draf 3 procedure.  

Was this decided prep or during surgery.   It is not clearly stated in the manuscript.”

Our apologies if it was not clear enough. The decision was made preoperatively. We added the following sentence in “Materials and Methods” section: “The specific type of Draf procedure was decided preoperatively, according to the duration and severity of symptoms, number and type of previous operations, presence of comorbidity (such as Sampters triad, allergy, asthma etc) and anatomical factors (anteroposterior and lateral size of frontal recess and presence of frontal cells)”  Draf classification is described in detail in “Introduction” section.

Reviewer 3 Report

This paper evaluates the questionnaire SNOT-22 before and after frontal sinus surgery in patients with CRS.

This paper does not describe the ethical considerations by the Ethics Review Board.

Abbreviations and References aren't considerably unified.

Surgical indications for Draf type IIb and III and disease details should be clearly shown.

The surgical procedure how they performed endoscopic sinus surgery is obscure.

Authors should investigate the correlations of the SNOT-22 with preoperative CT score and postoperative endoscopic findings.

Author Response

Dear reviewer,

Many thanks for going through our manuscript and for your helpful suggestions. We went through your suggestions and adapted our manuscript accordingly, as follows:

“Surgical indications for Draf type IIb and III and disease details should be clearly shown.

The surgical procedure how they performed endoscopic sinus surgery is obscure.”

Regarding indications, as stated before We added the following sentence in “Materials and Methods” section: “The specific type of Draf procedure was decided preoperatively, according to the duration and severity of symptoms, number and type of previous operations, presence of comorbidity (such as Sampters triad, allergy, asthma etc) and anatomical factors (anteroposterior and lateral size of frontal recess and presence of frontal cells)”  Draf classification is described in detail in “Introduction” section.

. Draf2a, Draf 2b and Draf 3 procedures and surgical indications are discussed in detail in “Introduction section”: ” The presence of cardinal symptoms of CRS (nasal obstruction, nasal discharge, face pain or pressure, hyposmia or anosmia) for more than 12 weeks with or without endoscopic and radiological confirmation and resistance to appropriate medical therapy are surgical indications4. When affecting the frontal sinuses, chronic sinusitis refractory to medical therapy is an indication frontal sinus surgery.

 Frontal sinus surgery is classified by Draf into types I, II and III, grading from simple drainage to endonasal median drainage. The type I drainage is established by ethmoidectomy including the cell septa in the region of the frontal recess and excluding the inferior part of Killian’s infundibulum and its mucosa. Resecting the floor of the frontal sinus between the lamina papyracea and the middle turbinate (type II a) or the nasal septum (type II b) anterior to the ventral margin of the olfactory fossa after ethmoidectomy enhances extended drainage. Type III enlarges the extended IIb opening by resecting the superior anterior nasal septum and drilling the floor of the frontal sinuses, between the orbits and up to first olfactory fibers 5. The choice of the appropriate procedure depends on the disease (recalcitrant, Samter’s triad), the anatomy (anteroposterior and lateral dimensions of frontal recess) and the patient (primary or revision surgery, other comorbidities,); however, the general principle of frontal sinus surgery applies: the aim is to create durable openings for sufficient drainage of frontal sinus and for improved delivery of topical therapies 6.” Also in “Materials and Methods” section we discuss again surgical indications: “All 99 patients suffered from CRS with or without nasal polyps and underwent frontal sinus surgery as a sole sinus procedure. The specific type of Draf procedure was decided preoperatively, according to the patient’s need for sufficient drainage of frontal sinuses. Patients listed for frontal sinus surgery were those resistant to appropriate medical treatment, as defined in ICARS 2016 guidelines 4

Authors should investigate the correlations of the SNOT-22 with preoperative CT score and postoperative endoscopic findings.”

Our intention was to emphasize on quality of life outcomes. Of course we included CT scan and nasal endoscopy in preoperative and postoperative evaluation of each patient, however, our aim was to compare SNOT-22 scores between Draf 2 and Draf 3 groups pre- and post-op.

“This paper does not describe the ethical considerations by the Ethics Review Board.”

For methods used in the study we applied for and received approval by the Ethics Review Board. We added a sentence in “Materials and Methods” section: “The methods used in the study were approved by the Ethics Review Board.”

Round 2

Reviewer 1 Report

All questions and comments have been addressed with exception of the subsequent one:

  • In Table 5 you report at 1-3 months follow-up a SNOT-22 mean score 22.87 for Draf2 and 37,00 for Draf 3. However, on Table 6 for the same period you report different SNOT-22 scores (22,8 for Draf 2 and 22 for Draf 3). Why? Also in line 126 you report a SNOT-22 score for Draf 2 as 22,7. Why? Please correct these discrepancies. Also report which is the follow-up time in line 126-128 (1-3 months post-op?).

Author Response

Dear reviewer, 

Many thanks for going through our revised manuscript and for your new helpful suggestions

Table 5 refers to mean changes in SNOT-22 total scores in Draf2 and Draf 3 group (22,87 and 37 respectively), whereas table 6 refers to mean SNOT-22 total scores (22,7 and 22 respectively). We also made clear that follow-up time in lines 125-126 was 1-3 months postoperatively, adding the following sentence:"There was no difference in SNOT 22 scores between the two groups postoperatively (1-3 months post-op): SNOT 22 total scores were 22,7 for the Draf 2 group and 22 for the Draf 3 group (p=0.8)." We believe that there are no more any discrepancies regarding scores.

With kind regards,

Andreas Liodakis

Reviewer 3 Report

They need the actual approval number by the Ethics Review Board at their hospital.

They did not correct my suggestion about abbreviations that should be spelled out at the first place in the text and format of References that is not unified.

For further scientific proof, we still recommend investigating the correlation between SNOT-22 and preoperative CT scores and postoperative endoscopic findings.

Author Response

Dear reviewer, 

Many thanks for going through our revised manuscript  and for your helpful suggestions.

  1. “They need the actual approval number by the Ethics Review Board at their hospital.”

 We requested and received the approval of the Scientific and Ethics Board of Hygeia Hospital  and also submitted as addendum the written approval.

 We also added in the Methods section the following statement

The scientific and ethics board of our hospital (chair: Dr G Zacharopoulos) approved this questionnaire based outcome study (approval letter included in addendum)”

  1. “They did not correct my suggestion about abbreviations that should be spelled out at the first place in the text and format of References that is not unified.”

 We have now spelled out all abbreviations as they appear in the text (in order of appearance Chronic Rhinosinusitis(CRS) / SNOT-22 (SinoNasal OutcomeTest–22) / patients with(CRSwNP) or without(CRSnNP) nasal polyps / quality of life (QOL) / Rhinosinusitis Outcomes Measure-31 (RSOM-31) / osteoplastic frontal sinus obliteration (OFSO) / Standard Deviation (SD)

The text has been updated to provide spelling for all abbreviations the first time they appear in the text, as follows:

Line 25 Chronic Rhinosinusitis(CRS)

Line 13 SNOT-22 (SinoNasalOutcomeTest–22) /

Lines 15 – 16 chronic rhinosinusitis patients with(CRSwNP) or without(CRSnNP) nasal polyps

Line 28 quality of life (QOL)

Line 48-49 Rhinosinusitis Outcomes Measure-31 (RSOM-31)”

Line 142 osteoplastic frontal sinus obliteration (OFSO)

Line 138 Standard Deviation (SD).

Line 402 - FESS (Functional Endoscopic Sinus Surgery)

We have further unified the format of references.

3. “For further scientific proof, we still recommend investigating the correlation between SNOT-22 and preoperative CT scores and postoperative endoscopic findings.”

As requested, we have done two additional studies in our data: we assessed the postoperative endoscopic findings (patency of Draf 3/ Draf 2 opening) at last followup, and recorded it as open, partly open or closed.

We adjusted introduction as follows:

The goal of this paper is to assess the effect of endoscopic frontal sinus surgery (Draf II and III) on health related quality of life, as recorded in pre- and postsurgical SNOT-22 questionnaires. A secondary goal was to assess the correlation between endoscopic findings (neo-ostium patency) and patient reported QOL as well as with radiological disease severity as recorded by Lund – MckKay grading scale.

We added this in the methods section as follows:

“We also recorded the postoperative status of the frontal sinus ostium / neo-ostium in all patients as follows: 0 – closed, 1- partly open, 2 – fully open.  In cases of bilateral Draf 2, we calculated the mean of both ostia.”

We then assessed the mean SNOT 22 score for these subgroups of patients as follows in our results section:

 At last follow up, in one Draf 2 patient the frontal ostium was closed, in 11 (15,5%) it was partly open and in 59 (83%) it was fully open.   In Draf 3 patients, one neostium was completely obstructed, 9 (31%) were partly open and 13 (56,5%) were completely open.  Snot 22 scorew were worse for patients where the ostium was closed (mean 43) or partly open (Mean 21,5) than those fully open (19) although the difference was not statistically significant. 

We added the Image 1: SNOT 22 mean and range by postoperative frontal sinus endoscopy

We also added in our discussion section the following statement and reference:

We did not find a statistically significant correlation between status of neoostium and SNOT 22, reflecting our previous study in Draf 3 (reference) although this could be related to the small number of patients with closed neo ostium. However, there was a trend towards worse QOL (from 43 to 20) in patients with closed neoostium.

Finally, we amended the abstract by adding the sentence:

Frontal sinus ostium was patent (fully or partly) at last follow up in 98% of patients after Draf 2 and in 88% of patients following Draf 3. Patients with non patent frontal (neo-) ostium had a mean postoperative SNOT 22 score of 43 compared to 20 of those with patent frontal sinus (neo-) ostium, although the difference was not statistically significant.

We also looked at the pre-operative frontal sinus radiological score and total L-M score of patients undergoing Draf 2 and Draf 3. As per L-M we graded each frontal sinus as 0 – completely clear, 1 – partly opacified, 2 – completely opacified.  We added this in our methods section as follows:

We recorded the preoperative computed tomography (CT) findings using Lund-Mckay  scale. As per L-M  we graded each frontal sinus as 0 – completely clear, 1 – partly opacified, 2 – completely opacified (total score for both frontal sinus 0-4 and total score for all sinuses 0-24).

We then assessed their correlation in our results section as follows:

All patients underwent CT preoperatively with a mean total L-M score of 14,5 (SD 5,4) and frontal sinus ra diological score of 2.46 (SD 1,3). There was no correlation between L-M score  or frontal sinus radiological scores and SNOT 22 scores (pearson correlation coefficient: 0,12, p=0.25 and 0,081 and p = 0.44 respectively).

And in our discussion section we added the following statement and reference:

In an earlier study (Correlation Between Symptoms and Radiological Findings in Patients With Chronic Rhinosinusitis: An Evaluation Study Using the Sinonasal Assessment Questionnaire and Lund-Mackay Grading System), we have shown that there is a poor correlation between QOL and radiological grading in this study we showed the same – although radiology is crucial for planning surgery and estimating the extent of disease, it is a poor predictor of QOL impairement – indeed, we found pearson correlation less than 0.2 between L-M score and preoperative SNOT 22.

Again, thank you for your help – we hope that we have improved our study based on your recommendations and it is now fit for publication.

With kind regards,

Andreas Liodakis